# Sex-Specific Effects of Long-Term Antipsychotic Drug Treatment on Adipocyte Tissue and the Crosstalk to Liver and Brain in Rats

**DOI:** 10.3390/ijms25042188

**Published:** 2024-02-11

**Authors:** Karin Fehsel, Marie-Luise Bouvier

**Affiliations:** Department of Psychiatry and Psychotherapy, Medical Faculty, Heinrich-Heine-University, Bergische Landstraße 2, 40629 Düsseldorf, Germany; bouvier@arcor.de

**Keywords:** perirenal adipose tissue, clozapine, haloperidol, lipid, glucose, perilipin-A hormone-sensitive lipase, glucose transporter-4, insulin receptor-ß

## Abstract

Antipsychotic drug (APD) medication can lead to metabolic dysfunctions and weight gain, which together increase morbidity and mortality. Metabolically active visceral adipose tissue (VAT) in particular plays a crucial role in the etiopathology of these metabolic dysregulations. Here, we studied the effect of 12 weeks of drug medication by daily oral feeding of clozapine and haloperidol on the perirenal fat tissue as part of VAT of male and female Sprague Dawley rats in the context of complex former investigations on brain, liver, and blood. Adipocyte area values were determined, as well as triglycerides, non-esterified fatty acids (NEFAs), glucose, glycogen, lactate, malondialdehyde equivalents, ferric iron and protein levels of Perilipin-A, hormone-sensitive-lipase (HSL), hepcidin, glucose transporter-4 (Glut-4) and insulin receptor-ß (IR-ß). We found increased adipocyte mass in males, with slightly higher adipocyte area values in both males and females under clozapine treatment. Triglycerides, NEFAs, glucose and oxidative stress in the medicated groups were unchanged or slightly decreased. In contrast to controls and haloperidol-medicated rats, perirenal adipocyte mass and serum leptin levels were not correlated under clozapine. Protein expressions of perilipin-A, Glut-4 and HSL were decreased under clozapine treatment. IR-ß expression changed sex-specifically in the clozapine-medicated groups associated with higher hepcidin levels in the perirenal adipose tissue of clozapine-treated females. Taken together, clozapine and haloperidol had a smaller effect than expected on perirenal adipose tissue. The perirenal adipose tissue shows only weak changes in lipid and glucose metabolism. The main changes can be seen in the proteins examined, and probably in their effect on liver metabolism.

## 1. Introduction

Weight gain under second-generation antipsychotic (SGA) medication associated with metabolic dysfunctions leads to increased morbidity and mortality with potential non-compliance in patients. The risk of weight gain and obesity appears to be the highest with olanzapine and clozapine [1]. An increase in weight has also been reported in individuals medicated with first-generation antipsychotics (FGAs) [2]. Especially adipose tissue as a target of SGAs seems to be involved in the etiopathology of obesity and metabolic abnormalities like type 2 diabetes (T2D) or insulin resistance, including glucose and lipid dysregulation [3]. In murine adipocytes, SGAs increase both adipogenesis of preadipocytes (hyperplasia) [4] and/or lipid accumulation (hypertrophy) in mature adipocytes [5].

Energy balance includes energy intake, energy expenditure and energy storage. When energy intake is equal to energy expenditure, body weight remains stable [6]. The key role of white adipose tissue (WAT) consists of the storage of energy and in the protection from excessive glucose in the form of triacylglycerols (TAG) and similarly named triglycerides (TG) in lipid droplets organelles. Perilipin is an intracellular protein that is hormonally regulated and is localized exclusively on the surface of lipid droplets. When energy is required, perilipin is phosphorylated by protein kinase-A. Once activated, perilipin changes its conformation and allows lipolysis by the three lipases adipocyte-triglyceride-lipase (ATGL), the rate-limiting enzyme for TAG hydrolysis, hormone-sensitive-lipase (HSL) and monoacylglycerol-lipase (MGL) which degrade TAGs sequentially, resulting in the release of free fatty acids and glycerol [7]. The fatty acids are then bound to albumin and transported to the target tissue. Glycerol can be isomerized in the liver to glyceraldehyde-3-phosphate and used for gluconeogenesis or degraded to pyruvate by glycolysis. Consequently, production of glycerol or lactate is an important primary function of adipocytes to lower the negative effects of excess glucose [8].

Increased VAT is associated with metabolic dysfunctions like T2D, insulin resistance, inflammatory processes and cardiometabolic disease [9]. In contrast to subcutaneous adipose tissue (SCAT), visceral preadipocytes originate from the mesothelium [10]. VAT is located around internal organs such as the uterus and ovaries in females (gonadal), epididymis and testes in males (epididymal), kidneys (perirenal), heart (epicardial) and intestines (retroperitoneal, mesenteric, and omental). Perirenal VAT has a close association to obesity and cardiovascular disease [11]. SCAT is located under the skin and serves primarily as a body temperature insulator and energy store. VAT adipocytes are more metabolically active, more sensitive to lipolysis and more insulin-resistant than SCAT adipocytes [12]. SGA therapy increases both subcutaneous and visceral adipose tissue in humans and rodents [3]. In recent years, VAT has been recognized as a major endocrine organ that secretes hormones like leptin, adiponectin, tumor necrosis factor-α (TNFα), interleukin-6 (IL-6) and estradiol, controlling whole body metabolism, insulin sensitivity, and energy homeostasis [13]. Leptin passes through the brain barrier and acts on neurons in the hypothalamic nuclei regulating food intake and energy metabolism by inhibiting the neuropeptides NPY (Neuropeptid Y) and agouti-related peptide (AGRP) [14]. It also induces lipolysis in adipocyte tissue, mediated by sympathetic neurons which innervate adipocytes in VAT [15]. Adiponectin is involved in the regulation of glucose levels as well as fatty acid oxidation. In humans and rodents, higher levels of the anti-inflammatory and anti-atherogenic adiponectin counteracts insulin resistance by reducing TG levels in muscle and liver cells [16]. However, it is also involved in the pathogenesis of obesity and non-alcoholic fatty liver disease (NAFLD), the hepatic manifestation of metabolic syndrome, characterized by increased hepatic lipid accumulation [17,18]. NAFLD covers a wide spectrum of disorders ranging from steatosis with a benign non-progressive clinical course, to finally advanced fibrosis and cirrhosis [19]. Ghrelin is mainly secreted by the gastric fundus mucosa and pancreatic cells. Circulating levels of ghrelin are increased in response to fasting or prolonged food restriction [20]. In addition to its stimulation of appetite and food intake by activating NPY and AGRP in the hypothalamus, ghrelin stimulates adipogenesis and suppresses lipolysis in rat visceral adipocytes [21]. The insulin-dependent glucose transporter-4 (Glut-4) is primarily expressed in adipocytes, skeletal and cardiac muscle cells. The binding of insulin to its receptor triggers kinase cascades that lead to translocation, docking and fusion of Glut-4 containing vesicles into the plasma membrane, allowing more glucose to enter the cell [22]. Through this glucose uptake by Glut-4 blood glucose levels can be lowered rapidly and efficiently, thereby reducing the harmful effect of excessive glucose levels.

Previously, we have examined the effects of clozapine and haloperidol on weight, food and water intake, and serum parameters [23], hepatic [24], hypothalamic [25] blood and iron parameters [26] in a rat model of metabolic syndrome. Finally, in this study, we examine the effects of the two antipsychotic drugs on basal metabolism of perirenal adipocytes. All tissues and data examined so far originate from the same animal trial.

## 2. Results

In short, male and female rats received daily treatment of clozapine or haloperidol by oral feeding in ground pellets over 12 weeks. Serum levels of haloperidol, clozapine, and its metabolite N-desmethylclozapine, body weight, motor activity, food and water intake (raw data and calculated on 1 kg body weight), and metabolic parameters were reported previously [23,24,25,26]. Male and female rats with haloperidol gained less weight than the control groups, associated with reduced food and water intake. Male rats under clozapine medication showed significantly increased body weight and liver mass, although their food and water intake were not increased. Higher fasting glucose in serum and decreased glycogen in liver were found in female rats treated with clozapine.

### 2.1. Effect of Haloperidol and Clozapine on Percentage Body Weight Gain after 12-Week Treatment of Clozapine and Haloperidol on Male and Female Sprague Dawley Rats

Percentage weight gain differed for TREATMENT [F(2,54) = 27.94 *p* = 0.0005] with significant effect between male controls and clozapine-medicated rats (*p* = 0.04) (Figure 1).

### 2.2. Effect of Haloperidol and Clozapine on Adipocyte Mass, Neutral Fat Content, Adipocyte Area Value, and Ferric Iron Content of Perirenal VAT

We found differences in adipocyte mass (Table 1) for TREATMENT [F(2,54) = 8.90 *p* = 0.00046], SEX [F(1,54) = 111.10 *p* ≤ 0.000001] and TREATMENTxSEX [F(2,54) = 5.41 *p* = 0.007]. Male clozapine-treated animals showed higher adipocyte mass than male controls (*p* = 0.047) and male haloperidol-treated animals (*p* = 0.001). Perirenal VAT of female haloperidol-treated rats differed from clozapine-treated animals (*p* = 0.020). Adipocyte mass of males differed significantly from females in all groups (*p* = 0.00006 for controls; *p* = 0.003 for haloperidol-medicated rats; *p* < 0.000001 for clozapine-medicated rats).

Neutral fat content in adipose tissue (Table 1) did not differ between the groups, but differed for SEX [F(1,54) = 111.10 *p* = 0.001], with differences between male and female controls (*p* = 0.021) and male and female haloperidol-treated animals (*p* = 0.035).

The mean adipocyte area value of 200 calculated perirenal adipocytes from each of two slices (Table 1 and Figure 2) differed for SEX [F(1,53) = 13.66 *p* = 0.00047], with differences between male and female controls (*p* = 0.041) and clozapine-treated rats (*p* = 0.018).

We did not find ferric iron deposits in adipose tissue, although the Perl’s Prussian blue stain is an extremely sensitive method.

Serum leptin levels are positively correlated with adipocyte mass in male (ρ = 0.830 *p* = 0.003) and female (ρ = 0.794 *p* = 0.006) controls, male (ρ = 0.733 *p* = 0.016) and female (ρ = 0.883 *p* = 0.002) haloperidol-treated animals, but not under clozapine treatment (Appendix A). Serum adiponectin levels were negatively correlated with percental adipocyte mass in the male (ρ = −0.661 *p* = 0.038) and positively in the female (ρ = 0.667 *p* = 0.050) haloperidol-treated group (Appendix A).

### 2.3. Effect of Haloperidol and Clozapine on Triglycerides, and NEFAs in Rat Perirenal Adipose Tissue as Well as Serum Cholesterol/HDL Ratio and LDL/HDL Ratio, and Hepatic Chol/HDL Ratio

The amount of triglycerides (Table 1) in perirenal adipocytes did not differ between the groups, but differed for SEX [F(1,51) = 5.01 *p* = 0.03] between male and female haloperidol-treated rats (*p* = 0.034). Hepatic and serum NEFAs (Table 1) did not differ between the groups, but in adipocytes for TREATMENT [F(2,51) = 3.38 *p* = 0.042] and SEX [F(1,51) = 8.37 *p* = 0.006] with decreased levels in male haloperidol-treated rats compared to male controls (*p* = 0.041). Only male haloperidol-treated animals differed significantly from the females (*p* = 0.002).

In the serum, LDL/HDL ratios (Table 1) differed for TREATMENT [F(2,52) = 12.71 *p* = 0.000032], and the Chol/HDL ratios (Table 1) for TREATMENT [F(1,52) = 15.07 *p* = 0.0000071], and SEX [F(1,52) = 13.39 *p* = 0.001]. Male clozapine-treated animals showed higher LDL/HDL and cholesterol/HDL ratios than male controls (LDL/HDL *p* = 0.011; cholesterol/HDL *p* = 0.007), and haloperidol-treated animals (LDL/HDL *p* = 0.002; cholesterol/HDL *p*= 0.001). Female clozapine-treated rats differed from female haloperidol-treated rats (LDL/HDL *p* = 0.032; cholesterol/HDL *p* = 0.025). The Chol/HDL ratio of male clozapine-treated rats differed significantly from females (*p* = 0.020).

### 2.4. Effect of Haloperidol and Clozapine on Glucose, Glycogen, and Lactate Contents in Rat Perirenal VAT

In perirenal adipose tissue glucose, glycogen and lactate did not differ significantly between the groups (Table 1). However, glycogen shows differences for SEX [F(1,53) = 5.77 *p* = 0.02], and TREATMENTxSEX [F(2,53) = 3.22 *p* = 0.048]. Male and female controls did not differ, but females under haloperidol (*p* = 0.023) and clozapine (*p* = 0.033) treatment had higher glycogen levels than males. We found no correlation between glucose and glycogen levels.

### 2.5. Effect of Haloperidol and Clozapine on Oxidative Stress of Rat Perirenal VAT and on Plasma Estrogen Level in Female Rats

We found no difference between the groups and for the sex of malondialdehyde equivalents in perirenal VAT (Table 1). Estrogen plasma levels did not differ between the female groups (Table 1).

### 2.6. Protein Expression of Perilipin-A, HSL, Hepcidin, Glut-4, IR-ß in Perirenal Adipocytes of Haloperidol- and Clozapine-Medicated Rat

The protein level of perilipin-A in adipocytes (Figure 3A) varied significantly for TREATMENT [F(2,51) = 5.91 *p* = 0.005], in which clozapine-treated females differed significantly from the control group (*p* = 0.006). Levels of HSL (Figure 3B) differed significantly for TREATMENT [F(2,53) = 14.45 *p* = 0.00001] and SEX [F(1,53) = 20.43 *p* = 0.000035]. Male clozapine-treated rats showed significantly lower levels than haloperidol-treated animals (*p* = 0.015), whereas females under clozapine treatment had lower levels than haloperidol-treated females (*p* = 0.001) and female controls (*p* = 0.012). All male groups had higher HSL levels than the respective female group (*p* = 0.032 for controls, *p* = 0.032 for the haloperidol-treated groups, *p* = 0.005 for the clozapine-treated groups). A positive correlation between perilipin-A and HSL was found in the male (ρ = 0.001) and a negative correlation in the female (ρ = −0.636 *p* = 0.048) control group (Appendix A). Hepcidin (Figure 3C) differed for TREATMENT [F(2,53) = 3.94 *p* = 0.025], SEX [F(1,53) = 5.50 *p* = 0.0023] and TREATMENTxSEX [F(2,53) = 3.90 *p* = 0.026]. Female clozapine-treated rats had higher hepcidin levels than female controls (*p* = 0.002) and haloperidol-treated females (*p* = 0.027). Only male controls had higher levels than female controls (*p* = 0.001). The corresponding autoradiographs of the stripped and re-probed membranes are presented in Appendix A in combination with the Ponceau S red stained membranes (Appendix A).

Examining Glut-4 (Figure 2D) with two-way ANOVA showed differences for TREATMENT [F(2,53) = 17.39 *p* = 0.000002], SEX [F(1,53) = 12.26 *p* = 0.001] and TREATMENTxSEX [F(2,53) = 6.94 *p* = 0.002]. Male haloperidol (*p* = 0.004) and male clozapine (*p* = 0.002) treated rats had significantly lower levels of Glut-4 than controls. Female clozapine (*p* = 0.001) treated rats had also lower levels, whereas female haloperidol (*p* = 0.001) treated animals showed higher expression of Glut-4 than clozapine-treated animals. Only male and female controls differed significantly from each other (*p* = 0.0003).

Two-way ANOVA of insulin receptor-ß (IR-ß) expression (Figure 2E) showed differences for TREATMENT [F(2,51) = 4.08 *p* = 0.023] and TREATMENTxSEX [F(2,51) = 16.23 *p* = 0.000004]. IR-ß was significantly increased in male clozapine-treated rats, compared to controls (*p* = 0.007), whereas females under clozapine treatment had lower levels than female controls (*p* = 0.009) and female haloperidol-treated rats (*p* = 0.00039). In clozapine-treated females, we found a nonparametric negative correlation between IR-ß and Glut-4 (ρ = −0.714 *p* = 0.047).

## 3. Discussion

One of the main findings of our study is that perirenal visceral adipose tissue (VAT)—at least in rats—does not significantly contribute to the weight changes under SGA or FGA treatment. In a previous publication [23], we were able to show that male Sprague Dawley rats under clozapine gained significantly more weight over the entire duration of the trial, while males under haloperidol significantly lost weight. Percentage weight also increases significantly in male rats under clozapine medication (Figure 1). Although perirenal adipocytes mass is significantly increased in the male clozapine-treated group (Table 1), adipocytes mass related to 1 kg body weight is not significantly elevated [23]. Neutral fat, triglycerides and NEFAs in perirenal adipocytes are not increased (Table 1). Liver mass also increases significantly in clozapine-treated males, as does the liver mass related to 1 kg body weight [24]. However, significant weight gain of male clozapine-treated rats or weight loss of haloperidol-medicated rats persists when fat mass and/or liver mass are subtracted from final body weight (Appendix A). Therefore, we conclude that there is at least one other parameter, such as subcutaneous adipose tissue (SCAT), that plays a role in weight gain with clozapine and weight loss with haloperidol. The sedative effect of haloperidol caused reduced food and water intake, and thus weight loss, in our animals [23]. In schizophrenic patients, medication with SGAs resulted in an increase in both VAT and SCAT [27]. Due to its promotion of inflammation, the accumulation of VAT is considered harmful, whereas the beige adipocytes in SCAT seem to have neuroprotective and anti-inflammatory effects [28]. Further studies need to clarify whether SCAT, which was neglected in our study because it is difficult to determine in rats, has a higher impact on weight gain or loss under antipsychotic drug medication than VAT. An answer to this question would certainly influence the clinical assessment of obesity in patients.

In murine cells, SGAs seem to increase both adipocyte number, due to proliferation and differentiation of progenitor cells in the adipose tissue (hyperplasia), and adipocyte size (hypertrophy), due to lipid accumulation in mature adipocytes [4]. Hypertrophy of adipocytes is associated with dysfunction of adipocytes and metabolic effects like altered lipid and energy metabolism, increased pro-inflammatory cytokine secretion, and insulin resistance [9,29]. In humans, hypertrophy of VAT but not SCAT is associated with a more atherogenic lipid profile, and several functional aspects of adipocyte activity relate to its size [9]. In our study, the increased adipocyte area value under clozapine points to hypertrophy (Table 1). Significant increases in leptin, triglycerides, insulin, intra-abdominal fat, adipocyte area and HSL activity are found in a metabolic syndrome model of male Wistar rats. This suggests that changed lipogenesis and lipolysis, hyperinsulinemia, and release of NEFAs cause a positive feedback loop that contributes to adipocyte hypertrophy [30]. But no direct influence of clozapine on fat storage and expression levels of an early marker of adipogenesis peroxisome-proliferator-activated-receptor-gamma (PPAR-γ), and a late marker of adipogenesis fatty-acid-binding-protein-4 (FABP-4) is found in human preadipocytes and mature adipocytes in vitro [31]. Unlike clozapine, which does not seem to have a significant effect on lipogenesis in mature adipocytes, haloperidol inhibits probably the lipogenic capacity of adipocytes in females, contributing to its lower propensity to induce weight gain [32].

Adipose tissue dysfunction as an intermediate between obesity and resulting inflammation and insulin resistance appears to be characterized by an increase in pro-inflammatory adipokines and a decrease in anti-inflammatory adipokines such as adiponectin [33]. Additionally, iron plays a role in the regulation of adipose tissue function, especially in the expression of adipokines [34]. In contrast to the liver [24], we do not find ferric iron deposits in adipocytes. Hepatic ferric iron accumulation is associated with oxidative stress, which is increased in liver [24] but not in adipose tissue (Table 1). Low cellular iron level leads to obese-resistant white adipose tissue and limits lipid uptake in enterocytes, reducing hypertrophy, lipid peroxidation and inflammation of adipose tissue, thereby reducing the development of hepatic steatosis and insulin resistance [35]. Hepcidin is considered the key regulator of iron metabolism. Haloperidol or clozapine-medicated female rats have higher serum ferritin levels and higher hepcidin levels in liver [26] and adipose tissue (Figure 3C). In mouse models of obesity, increased adiposity leads to iron deposits only in the epididymal adipose tissue (eAT) of male mice. This tissue-specific iron accumulation causes local adipose tissue insulin resistance, macrophage accumulation, and finally inflammation [36].

Clozapine might play an important role in the regulation of the adipokines leptin and adiponectin, and total ghrelin [37]. In healthy individuals, leptin and total body adipose mass are positively correlated [38], but we find this correlation for leptin and perirenal VAT only in the control and the haloperidol-treated groups (Appendix A). A negative correlation between obesity and circulating adiponectin is also well-established [9], and adiponectin concentrations increase concomitantly with weight loss, whereas decreased adiponectin levels are associated with insulin resistance and hyperinsulinemia [39]. In our study, we found an increase of serum adiponectin levels in all drug-treated groups, highest in the two clozapine groups [23] and, like expected, a negative correlation between serum adiponectin and perirenal VAT for male haloperidol-treated rats (Appendix A). After binding to its receptors, Adipo-R1 and R2, adiponectin initiates the phosphorylation of adenosine-monophosphate-activated-protein-kinase (AMPK), a key regulator of hepatic energy and lipid homeostasis [40]. Activation of AMPK suppresses fatty acid and cholesterol biosynthesis by inhibiting the activity of acetyl-CoA-carboxylase and 3-hydroxy-3-methyl-glutaryl-CoA-reductase. Furthermore, activation of AMPK inactivates the SREBPs, the transcriptional regulators of lipogenesis, thereby inhibiting lipogenic genes. Clozapine treatment suppresses AMPK-activity in male SD rats [41], thereby allowing upregulation of SREBP-1c in male and SREBP-2 in female rats [24]. SREBP-1c stimulates hepatic triglyceride accumulation in males [24], and SREBP-2 induces serum cholesterol production in females and partially in males [23]. Haloperidol has no effect on the SREBPs [24], and may inhibit cholesterol biosynthesis [42]. In healthy animals, glycogen-bound subunit AMPK-ß suppresses activation of AMPK [43]. Reducing the glycogen stores allows AMPK to inhibit lipogenesis and to direct cell metabolism towards fatty acid oxidation [44]. Downregulation of glycogen synthesis—probably by GSK3-ß activation—subsequently leads to hyperglycemia in female clozapine-treated rats [23]. In male rats treated with clozapine, the glycogen stores are only slightly reduced. The both atherogenic serum indices cholesterol/HDL and LDLl/HDL ratios (Table 1) are increased significantly in the male and weakly in the female clozapine-treated groups. The same is seen in schizophrenic patients [45]. Therefore, we hypothesize that the increase of adiponectin levels counteracts cholesterol-regulated lipogenesis by stimulating the hepatic adiponectin receptors, which activate AMPK as seen in C2C12 myotube cells [46]. AMPK activation induces peroxisome proliferator-activated-receptor-α (PPAR-α) signalling, which in turn induces anti-inflammatory signalling, reduces adipose-tissue-derived factors that could stimulate steatosis [18], and inhibits SREBP activity, thereby reducing the development of hepatic steatosis [47]. Activation of PPAR-α causes a reduction in circulating triglycerides and an increase in the uptake of free fatty acids. Surprisingly, chronic treatment with clozapine or haloperidol lowers adipocyte NEFAs in the male-treated groups, as seen under olanzapine medication in mice and humans, but not in serum or liver (Table 1), whereby insulin resistance, adiposity, and diabetes are typically associated with elevated free fatty acids [48]. Ghrelin is elevated during fasting periods and decreases rapidly after food intake. Ghrelin not only promotes adiposity by activating hypothalamic orexigenic neurons, but also directly stimulates fat storage-related proteins in adipocytes like Acetyl-CoA-carboxylase, fatty-acid-synthase, lipoprotein-lipase and perilipin, thereby stimulating intra-cytoplasmic lipid accumulation [49]. In humans, perilipin is elevated in obese subjects and its level is enhanced in steatotic hepatocytes [50]. In our study, the protein expression of perilipin-A and HSL in adipose tissue is decreased under clozapine treatment in both sexes (Figure 3A,B), indicating no effect of the antipsychotic drug on lipolysis and lipogenesis. Therefore, we suspect that the serum ghrelin levels are elevated due to the 12 h starvation period of the rats before blood and organ collection.

In adipocytes, we find slightly decreased glucose in all treated groups, decreased glycogen in males and increased glycogen in females (Table 1) associated with decreased protein levels of the insulin-dependent Glut-4 in the clozapine-treated groups. Under conditions of low insulin, most Glut-4 is sequestered in intracellular vesicles in muscle cells and adipocytes. Increasing insulin levels, induced by increasing blood glucose, cause the uptake of glucose into the adipocytes by Glut-4, which is quickly incorporated into the plasma membrane of the cell when insulin binds to its membrane receptors. Thereby, insulin-stimulated glucose uptake enhances glycolysis, tricarboxylic-acid-cycle-flux and oxidative phosphorylation with mitoROS (reactive oxygen species) production in a glucose-dependent manner. Downregulation of insulin receptors, as well as insulin resistance, protect against glucose oversupply [51]. Haloperidol slightly increases serum insulin levels [23] associated with an increase of IR-ß in both sexes and a decrease of Glut-4 in adipose tissue of males. Clozapine treatment leads to a lower protein level of Glut-4 in males and females associated with an increase in IR-ß in males and a decrease in females (Figure 2D,E). Under normal conditions, the insulin level should rise dramatically—at least in females under clozapine treatment—due to the excessively elevated serum glucose level, with the result that more glucose would be taken up via Glut-4 into the adipocytes and would be stored there as triglycerides. However, despite strongly increased serum glucose levels, no more insulin is secreted. The fact that NEFAs are not elevated in any of the tissues examined also argues against a classic insulin resistance (Table 1). Murashita et al. [52] observed elevated blood glucose levels also in male clozapine-treated SD rats. Clozapine directly inhibits insulin secretion through membrane depolarization and activation of protein kinase C in pancreatic cells [53]. Smith et al. [54] found that SGA treatment in rats can interfere with glucose metabolism, not by acutely inducing insulin resistance, but by increasing pancreatic glucagon secretion, and thus stimulating hepatic glucose production. Hepatic glucose output (HGO) derives from glycogen breakdown (glycogenolysis) and/or from de novo synthesis of glucose (gluconeogenesis). We found liver glycogenolysis in females under clozapine [24], and the significantly increased hepatic pyruvate and lactate levels in males favor suspected gluconeogenesis in males (Table 1). Glut-4 expression in insulin-sensitive tissues decreases in association with whole-body insulin resistance and high levels of inflammatory markers in a rat model of metabolic syndrome [55]. Clozapine is known to stimulate proinflammatory gene transcription in human adipocytes [56]. Studies implicate a complex network by which glucose sensing through Glut-4 in muscle and adipocytes may operate to integrate whole-body energy metabolism [57], and adipocyte insulin sensitivity appears critical for whole-body homeostasis.

We have investigated the impact of clozapine and haloperidol on metabolically active tissues such as liver and VAT, and their neuronal control in the hypothalamus, the control center of body weight and homeostasis. It has been shown that the affected organs do not act independently of each other, but rather influence each other in a sex-specific manner (Figure 4). Under clozapine treatment, increased hypothalamic lactate level in females points to enhanced glycolysis with a higher glucose demand to produce the required energy. The two insulin-independent glucose transporters Glut-1 and Glut-3 were upregulated, and this refers to glucose deficiency in endothelial cells, and ultimately in neurons and astrocytes. We found excessive fasting glucose in the serum of female rats treated with clozapine. The glucose probably originates from the hepatic glycogen depot, which was either no longer replenished or had been emptied. Glycogenolysis in female liver contrasts with the findings in male rats, which compensate their hypothalamic glucose deficiency through gluconeogenesis, and carry out increased glycolysis with high hepatic pyruvate and lactate values. In contrast to VAT, the liver shows high oxidative stress for both sexes. In females, hemosiderin is deposited in the liver, while males show an iron deficiency, which is reflected in a low hemoglobin content and a reduced cytochrome level [26]. Males have extremely high hepatic triglyceride levels and increased lipid droplets. This is consistent with the upregulation of SREBP-1c in males and SREBP-2 in females, which have high serum cholesterol, HDL-cholesterol and LDL-cholesterol levels. Based on high triglyceride values, high NEFAs levels, increased liver mass and the accumulation of lipid droplets, we assume the onset of steatosis in male rats under clozapine.

Unexpectedly, adipose tissue shows a much less disturbed lipid and glucose homeostasis without oxidative stress than the liver [24]. Low cellular iron levels limit the absorption of lipids, thereby reducing hypertrophy, reduce lipid peroxidation and inflammation and decrease the development of steatosis and insulin resistance in the liver [26]. Clozapine suppresses AMPK-activity and upregulates SREBP-1c in males [24], leading to hepatic triglyceride accumulation, and SREBP-2 in females [24] increasing the cholesterol level in serum [23]. The increased release of adiponectin [23] counteracts this effect and activates the hepatic AMPK, and reduces fatty acid and cholesterol biosynthesis. In addition, AMPK can inhibit lipogenesis by depleting glycogen depots, as we were able to show in female rats [24]. Clozapine does not cause classic insulin resistance in our animals. Despite the very high serum glucose level in female animals [23], insulin secretion did not raise suggesting a non-classic insulin resistance.

### Limitations

The limitations of our study, such as the social isolation of the test animals or the small group size, have already been discussed in the four previous publications [23,24,25,26]. Among the major limitations of our study on adipose tissue, is the fact that we only studied perirenal VAT, although this only accounts for 20% of the total adipose tissue [9]. Clozapine has no acute effect on food intake in rats [23,58], unlike in patients, in whom, however, no precise studies exist on the amount of food ingested and the amount of energy consumed under clozapine or haloperidol. The cells of smaller mammals such as rats or mice consume more oxygen and nutrients per unit time. This, as well as the metabolic differences between humans and rats, leads to the assumption that the results from our animal model can only be transferred to humans to a limited extent and do not always mimic clinical findings [59], although rodent adipose tissue shows great similarity to human adipose tissue [60]. Nevertheless, the results of our study show many similarities with the clinical findings under antipsychotic medication, and some results can only be obtained in an animal model. In our studies, male and female animals were tested. It has been shown that there are many sex-related differences. Increasing clinical research shows also that male and female patients differ in their metabolic expression of medication [61].

## 4. Methods

### 4.1. Ethics Statement

All experiments were carried out in accordance with the laws of the local authorities for animal experiments approved by the Landesamt für Natur, Umwelt- und Verbraucherschutz NRW, Recklinghausen, Germany; Reference number 9.93.2.10.34.07.227. The feeding experiment took place in the rooms of the animal testing facility of the Heinrich Heine University, Düsseldorf. The study was conducted in compliance with ARRIVE guidelines 2.0. The animal husbandry before and during the feeding experiment, as well as the killing, took place in the rooms of the animal experiment facility (TVA) of the Heinrich Heine University, Düsseldorf under the control of Dr. Treiber, director of the TVA.

### 4.2. Experimental Design

In total, 30 male and 30 female healthy 10 week old Sprague Dawley rats from Taconic (Denmark) housed in 2 × 5 groups of 6 animals (sibling group), separated by sex, from the same litter with free access to water and ground food pellets (maintenance diet (Altromin Spezialfutter GmbH, Lage, Germany) with 19% crude protein, 4% crude fat and, additionally, 15% fat to facilitate drug uptake). The animals were maintained on a 12:12 h light/dark cycle at 21 °C and 60% humidity. For acclimatization, the animals were handled every day and weighted 2 times per week from postnatal day (PD) 71. On PD 78 the rats were separated from each other and individually housed during the test period, and were divided into the control, haloperidol-medicated and clozapine-treated groups, resulting in 10 animals per group (*n* = 10). To avoid puberty effects, antipsychotic drug treatment started at PD 85 (week 13) to PD 165 (week 25). Male (*n* = 10 per group) or female (*n* = 10 per group) rats were fed orally each day with clozapine (Leponex^®^, Novartis, Cologne, Germany with 20 mg/1kg body weight (BW)/day in ground pellets), corresponding to an effective average dose rate to 18.5 + 0.26 mg/kg BW for males and 17.7 + 0.38 mg/kg BW for females [23]. Haloperidol (Haloneurol^®^, Hexal, Holzkirchen, Germany) was fed with 1mg/1kg BW/day in ground pellets, corresponding to an effective daily average dose of 0.8 ± 0.03 mg/kg BW for males and 0.6 ± 0.08 mg/kg BW for females [23]. Results were compared to those in male and female control groups fed only with ground pellets. On PD 169 (week 25), 12 h after food removal, the animals were anaesthetized by Pentobarbital (Narcoren, Merial, Ingelheim, Germany), and perirenal fat pads as part of the intra-abdominal VAT were removed, weighted, and stored at −80 °C in an ultra-low temperature cooler. All metabolic parameters were determined for all test animals.

### 4.3. Determination of Percentage Weight, Neutral Fat Content and Adipocyte Area Value

To calculate the percentage weight, we took the difference between the final and initial weight and calculated its percentage of total weight.

A total of 0.5 g adipose tissue was homogenized on ice in 1 mL RIPA buffer (50 mM Tris-HCl, pH 7.4, 150 mM NaCl, 1% Na-deoxycholate, 0.1% LDS, 1 mM EDTA, 3% Triton with protease inhibitor (complete mini, EDTA free, Roche Diagnostics, Mannheim, Germany) overnight at 4 °C and centrifuged at 1000× *g* for 10 min at 4 °C. The pellet, containing blood vessels and collagen, was discarded and the aqueous supernatant was collected for measurement of triglycerides, NEFAS, glucose, glycogen and lactate. The fat phase was dissolved by shaking with 1 mL diethyl ether overnight at 39.5 °C. The liquid phase was transferred in weighted tubes, the volatile diethyl ether was removed, and the weight of neutral fat was determined.

The adipocyte area value was determined microscopically. A total of 20 µm sections of frozen perirenal VAT were fixed with 4% buffered neutral formalin at room temperature for 5 min, washed three times with distilled water and air-dried overnight. The slices were stained in Mayers hematoxylin (Carl Roth, Karlsruhe, Germany) for 5 min, rinsed in cool running water for 5 min, stained in 0.5% aquaeous eosin g solution (Roth, Germany), rinsed with distilled water, dehydrated in Roti-Histol (Roth, Germany), and mounted in Roti-Histokitt (Roth, Germany). Two clusters on two different sections of each animal, each cluster enclosing 200 adipocytes, were marked by x, and the two areas were calculated with Axiovision (Zeiss, Oberkochen, Germany) and averaged.

### 4.4. Determination of Ferric Iron by Perl’s Prussian Blue Staining of Frozen Fat Tissue Sections

We analyzed ferric iron (Fe^3+^) by Perl’s Prussian blue staining [62]. A total of 30 µm sections of frozen adipose tissue were fixed with 4% buffered neutral formalin at room temperature for 5 min, washed three times with distilled water, and air-dried. After descending alcohol series, the tissue was pre-stained in 10% potassium-ferrocyanide for 5 min, and stained with equal parts of a mixture of 2% potassium-ferrocyanide solution and 1% hydrochloric acid for 30 min at 37 °C. After washing with distilled water, the slides were counterstained with nuclear fast red aluminum sulphate solution (Roth, Germany) for 5 min, rinsed with distilled water, dehydrated in Roti-Histol (Roth, Germany), and mounted in Roti-Histokitt (Carl Roth, Germany).

### 4.5. Determination of Triglycerides and Non-Esterified Fatty Acids (NEFAs) in Adipocytes, Cholesterol/HDL-Cholesterol Ratio in Liver and Serum, and LDL-Cholesterol/HDL-Cholesterol Ratio in Serum

Triglyceride levels were determined by automated counter in the clinical laboratory of the LVR Klinikum Düsseldorf. NEFAs were determined in serum, liver, and adipocyte lysate by NEFA-HR (2) assay kit (436-91995, Fujifilm, Germany) following the manufacturer’s recommendations. The two atherogenic indices cholesterol/HDL-cholesterol ratio and LDL-cholesterol/HDL-cholesterol ratio were calculated from the values for cholesterol, HDL-cholesterol and LDL-cholesterol in the serum and cholesterol/HDL-cholesterol ratio in liver.

### 4.6. Determination of Glucose, Glycogen, and Lactate in Lysates of Fat Tissue

Glucose levels in adipocyte lysate were determined in the whole cell lysate by glucose colorimetric assay kit (MAK263, Sigma-Aldrich, Taufkirchen, Germany) following the manufacturer’s recommendations. Lactate was determined in the whole cell lysate by a lactate LOX-PAP test (LT-LC0056, Labor+Technik, Eberhard Lehmann GmbH, Berlin, Germany) following the manufacturer’s recommendations.

Glycogen content was determined by the anthrone method. A total of 0.25 g adipose tissue was mixed with 1.25 mL HClO_4_ (6%, ice cooled) and heated for 10 min at 80 °C. The solution was centrifuged for 5 min at 960 rpm, and 4 °C. 2.5 mL methanol was added to 1.0 mL supernatant, heated for 30 min at 37 °C, and centrifuged for 10 min at 8000 rpm and room temperature. The pellet was dissolved in 200 µL aqua dest., mixed with 400 µL anthrone solution (0.2% anthrone (Sigma Aldrich, Germany) in concentrated H_2_SO_4_), and heated for 15 min at 95 °C_._ After cooling to room temperature, the extinction of the green solution could be measured at 620 nm, and the levels of glycogen were calculated in mg/g of fat tissue using a standard curve.

### 4.7. Determination of Oxidative Stress in Perirenal Fat of All Rats and of Estrogen Levels in Serum

Lipid peroxidation in the adipose tissue was determined through the production of TBARS, as previously described by [63]. A total of 50 µL SDS (8.1%), 375 µL acetic acid (20%, adjusted to pH 3.5 with 1 N NaOH) and 375 µL TBA (0.8% aqueous solution of thiobarbituric acid) were added to 50 µL cell lysate and filled-up to 1 mL with aqua dest. The solution was boiled for 60 min and cooled down to room temperature. A total of 0.5 mL n-butanol + pyridine (15:1) were added, thoroughly mixed, and centrifuged for 10 min at 2000 rpm at room temperature. Spectral absorption was measured at 532 nm and malondialdehyde equivalents were calculated in µmol/mL by a standard curve.

Estrogen levels were measured in female rat serum with a rat estrogen ELISA kit (EKR3010, Nordic Biosite, Eching, Germany) following the manufacturer’s recommendations.

### 4.8. Western Blot Analysis of Adipocyte Lysates of the Drug-Medicated Rats and Controls

Protein expression was identified by Western blot analysis in the whole cell lysate. A total of 0.5 g fat tissue was homogenized on ice in 1.0 mL tissue extraction buffer I (50 mM Tris, pH 7.4, 250 mM NaCl, 5 mM EDTA, 2 mM Na_3_VO_4_, 1 mM NaF, 20 mM Na_4_P_2_O_7_, Thermo Fisher scientific, Dreieich, Germany) with protease inhibitor cocktail (complete mini, Roche, Germany) shaken overnight at 4 °C, and centrifuged at 13,000× *g* for 20 min at 4 °C. Total protein content was measured using the DC Protein Assay from Bio Rad (Feldkirchen, Germany). A total of 20 µg protein from each animal was separated on NuPAGE^TM^ 4–12% Bis-Tris gels (Thermo Fisher scientific, Germany) and blotted onto Invitrolon membranes (Thermo Fisher scientific). Membranes were then blocked with Roti Block (Carl Roth, Karlsruhe, Germany) in TBS-T (0.5:10) at RT for 30 min. The blocked PVDF membranes were incubated overnight at 4 °C either with rabbit polyclonal perilipin A antibody (62 kilodalton (kDa), 1:1000, PA1-1051, Thermo Fisher Scientific, Germany), rabbit polyclonal HSL antibody (80 kDa, 1:1000, PA1-16966, Thermo Fisher Scientific, Germany), goat polyclonal hepcidin antibody (9 kDa; 1:1000, sc-240553, Santa Cruz Biotechnology, Heidelberg, Germany), rabbit polyclonal Glut-4 antibody (54 kDa, 1:1000, ab216661, Abcam, Germany), and mouse monoclonal IR-ß antibody (95 kDa, 1:1000, sc-373975, Santa Cruz Biotechnology, Germany) in TBS-T, and then probed at room temperature for 1 h with the HRP-conjugated secondary antibody (against rabbit 1:20,000, ab97051, Abcam, mouse 1:20,000, sc-2004, Santa Cruz, or goat 1:20,000, sc-2064, Santa Cruz, respectively). Resulting autoradiographs by autoradiography films (Santa Cruz Biotechnology, Germany) (Appendix A) were measured by densitometry using Gene Snap software version 6.00 (q) and Gene Tools software version 3.002 (a) (Syngene, Synoptics Ltd., Cambridge, UK). The membranes were routinely stained with Ponceau-S (Appendix A) before immunostaining to confirm correct sample loading and the transfer of equivalent amounts of protein [64].

### 4.9. Statistical Analysis

Our study is an exploratory study. Therefore, we chose 10 animals per group (male and female controls, male and female haloperidol-medicated rats, and male and female clozapine-medicated animals) to obtain statistically valid conclusions.

We have re-analyzed the following previously published results [23,24]: final body weight of male clozapine-treated animals; data of serum leptin, adiponectin, ghrelin; and serum and liver cholesterol, HDL-cholesterol and LDL-cholesterol.

Statistical analysis was performed using IBM SPSS version 25 for Windows. All data were represented as mean ± SE. The data were examined by two-way ANOVA with the between-subject factors TREATMENT and SEX and one-way ANOVA with TREATMENT as factor. In case of significant group effects appropriate post hoc tests were carried out between the groups. Sex-dependent effects were examined by t-tests for independent samples. Due to the small number of animals per group and the explorative character of the study, *p*-values ≤ 0.05 were considered as statistically significant, and multiple testing was presented without α-adjustment. A bivariate correlation procedure with Spearman-Rho coefficient was carried out to test the relationship between HSL and perilipin-A, between serum adiponectin, leptin and body fat mass related to 1 kg BW, ghrelin and perilipin-A, IR-ß and Glut-4 with *p* ≤ 0.05 as significant. Significant correlations are displayed as simple scatterplots performed by spss in the Appendix A.

## 5. Conclusions

The examination of the perirenal adipose tissue shows lower metabolic changes under haloperidol and clozapine than in serum and especially in liver. The differences between male and female animals were also smaller in adipocytes (Table 1). Lactate or pyruvate levels in liver and hypothalamus of male rats indicate a shift to glycolysis under clozapine, but this is also less pronounced in adipose tissue, probably because adipocyte metabolism depends on oxidative phosphorylation. Under clozapine treatment, glucose seems to be stored in the male liver as triglyceride deposits, rather than in adipose tissue. Microscopic examination of adipose cells indicates slight hypertrophy, but without increased food intake [23]. However, the increase in adipose and liver tissue does not explain total weight gain, suggesting an additive involvement of subcutaneous adipose tissue. Increased protein expression of the IR-ß and decreased Glut-4 expression indicate insulin resistance under haloperidol and in clozapine-treated males, but not in females under clozapine treatment. Lipolysis or lipogenesis seem to be unaffected in adipocytes, while liver shows adipogenic transformation in male clozapine-treated rats leading to steatosis. Clozapine induces hepatic SREBP-1c expression in male rats [24], the upstream regulator of fatty-acid-synthase. Female rats are protected against fatty liver by low glycogen levels. Downregulation of glycogen synthesis and upregulation of glycogenolysis—probably by GSK3-ß activation—subsequently leads to higher serum glucose levels in female clozapine-treated rats. It is still an open question whether the glucose deficit of the neuronal cells in the hypothalamus [25] or the glycogen depletion in liver cells is responsible for the wide range of peripheral metabolic changes. The disturbance of glucose and lipid metabolism is tissue- and sex-dependent, and obesity shows sexual dimorphism, as seen in Palmer and Clegg [61].

## Figures and Tables

**Figure 1 ijms-25-02188-f001:**
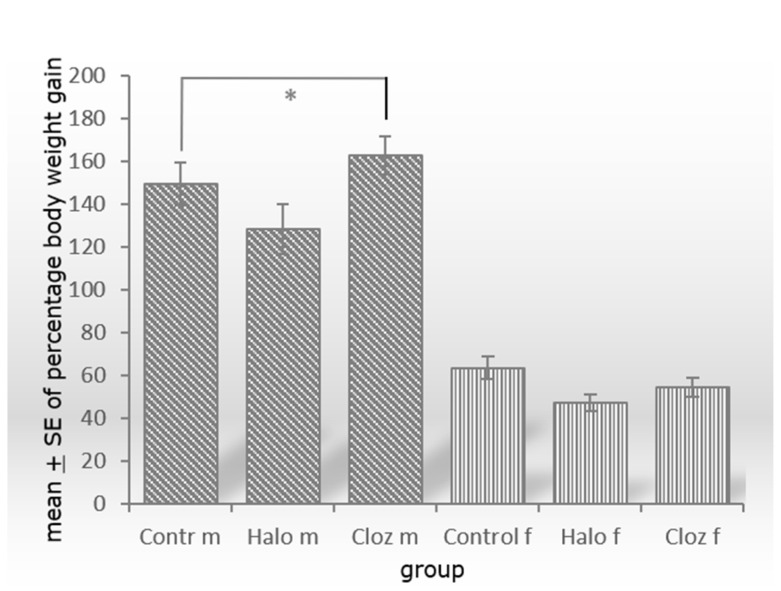
Significant increase of percental weight gain between male controls and clozapine-treated Sprague Dawley rats with *p* = 0.022 and *n* = 10 animals per group (m = male, f = female, Contr = Control, Halo = haloperidol-medicated, Cloz = clozapine-medicated). * *p* ≤ 0.05.

**Figure 2 ijms-25-02188-f002:**
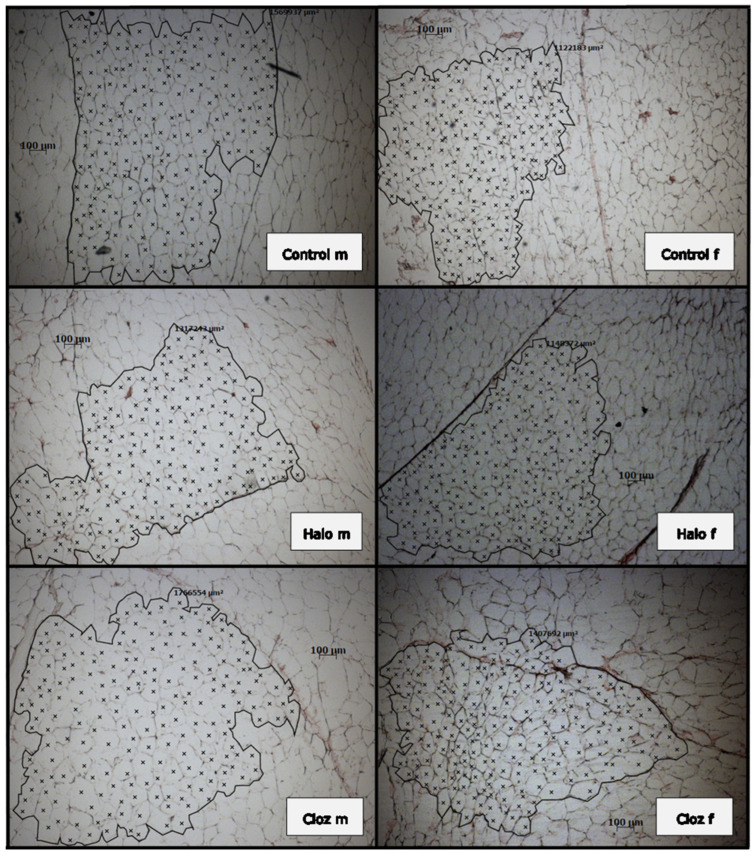
Exemplary presentation of 20 µm frozen fat sections of male and female Sprague Dawley rats after 12-week medication with haloperidol or clozapine, stained with Mayer’s hematoxylin. Two hundred adipocytes from each of two slices were counted (marked by X), their area was calculated with Axiovision (Zeiss, Germany) and the values were averaged. Values are given in Table 1 as “adipocytes area value”. (m = male, f = female, Control = controls, Halo = haloperidol-medicated, Cloz = clozapine-medicated).

**Figure 3 ijms-25-02188-f003:**
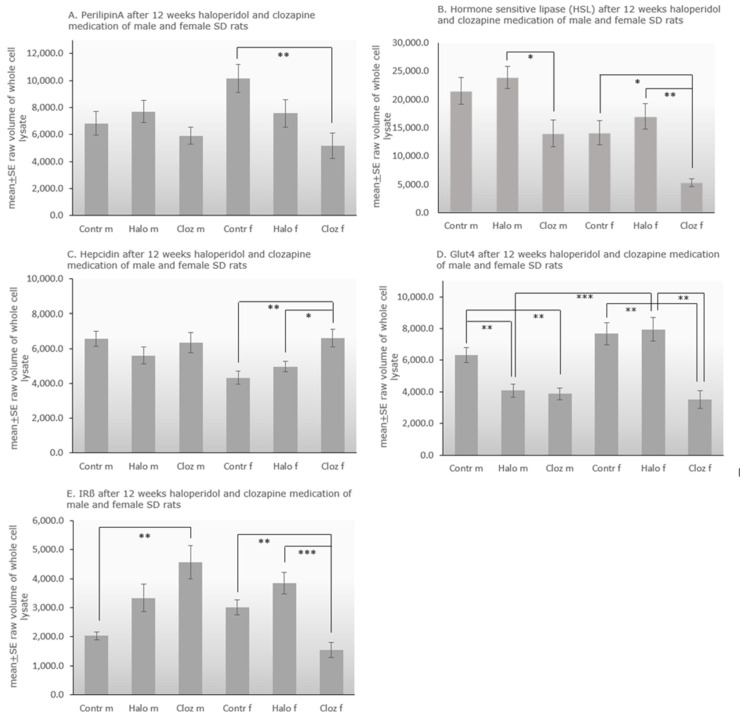
Protein expression of (**A**) perilipin A, (**B**) hormone-sensitive lipase (HSL), (**C**) hepcidin, (**D**) glucose transporter 4 (Glut 4), (**E**) insulin receptor ß (IR ß) in male and female control, haloperidol, and clozapine-treated Sprague Dawley rats with * *p* ≤ 0.05, ** *p* ≤ 0.001 and *** *p* ≤ 0.0001 as significant (*n* = 10 animals per group). The autoradiographs and the Ponceau-stained membranes are seen in Appendix A.

**Figure 4 ijms-25-02188-f004:**
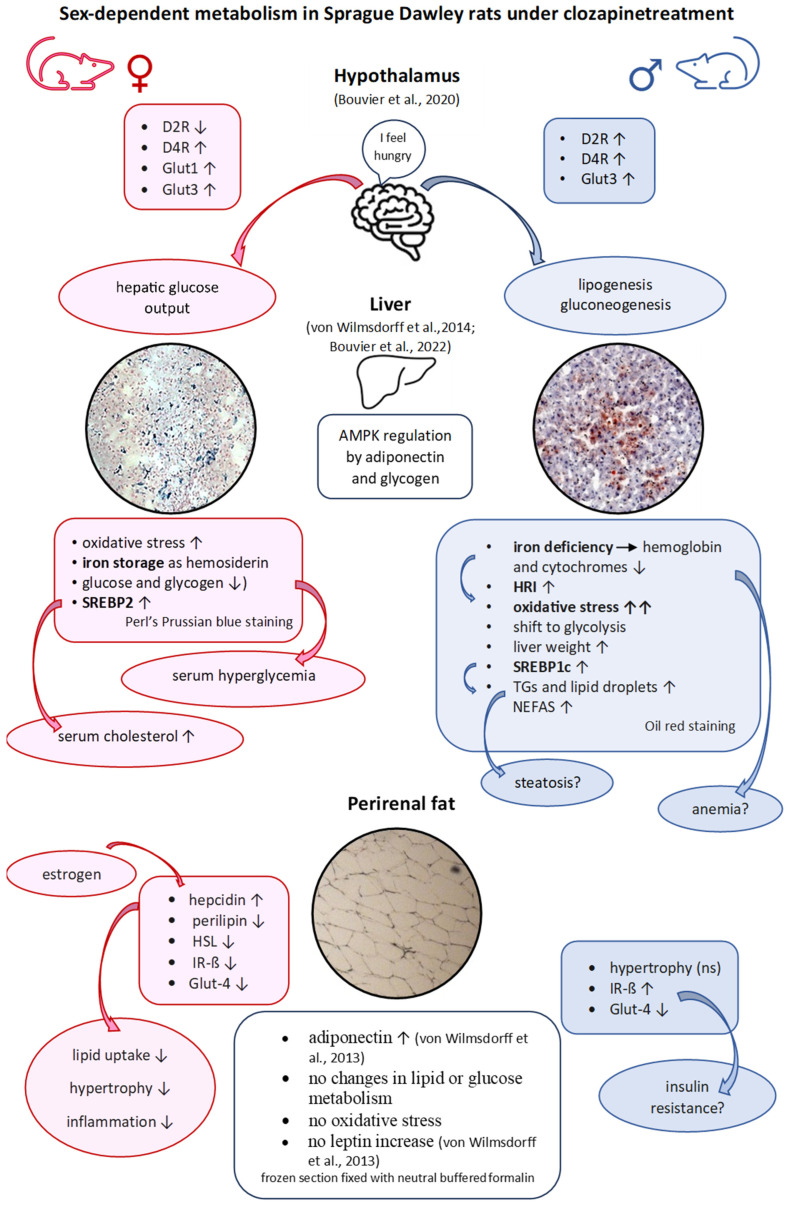
Schematic diagram of the organ connections (crosstalk) derived from the results of the serum [23,24], liver [24,26], hypothalamus [25], and adipose tissue examinations under clozapine medication. The explanations can be found in the text below.

**Table 1 ijms-25-02188-t001:** Main metabolic parameters of perirenal VAT and selected metabolic parameters of liver and serum in male and female control, haloperidol- and clozapine-medicated Sprague Dawley rats after 12-week trial period.

		Contr m	Halo m	Cloz m	Contr f	Halo f	Cloz f
adipocytes	percentage weight gain	149.9 ± 10.0	128.6 ± 11.7	162.8 ± 8.9 *	63.6 ± 5.3	47.3 ± 3.8	54.6 ± 4.4
	adipocytes mass [g]	27.55 ± 2.38	22.73 ± 2.60	36.13 ± 1.89 *	13.29 ± 1.34	12.35 ± 1.07	14.07 ± 0.81
	neutral fat content [g/g fat tissue]	0.51 ± 0.04	0.58 ± 0.05	0.43 ± 0.02	0.33 ± 0.06	0.41 ± 0.04	0.36 ± 0.04
	adipocytes area value [µm^2^]	1,475,632 ± 100,845	1,342,769 ± 86,813	1,635,892 ± 88,177	1,115,800 ± 62,396	1,109,041 ± 63,796	1,230,149 ± 74,687
	triglycerides [mg/dl]	236.7 ± 19.5	190.0 ± 11.2	222.9 ± 19.8	268.6 ± 19.9	258.8 ± 26.6	237.6 ± 26.7
	NEFAs [mmol/L]	0.54 ± 0.04	0.41 ± 0.01 *	0.47 ± 0.04	0.55 ± 0.02	0.52 ± 0.03	0.56 ± 0.04
	glucose [ng/µL]	15.71 ± 2.2	12.13 ± 2.4	11.89 ± 2.0	13.99 ± 1.9	12.69 ± 1.6	12.52 ± 2.3
	glycogen [µg/mL]	7.62 ± 1.36	4.80 ± 0.70	6.51 ± 0.78	6.43 ± 1.36	8.68 ± 1.40	11.31 ± 1.82
	lactate [mmol/L]	0.69 ± 0.04	0.85 ± 0.19	0.64 ± 0.04	0.73 ± 0.05	0.86 ± 0.15	0.89 ± 0.09
	malondialdehyde [nm/mL]	32.72 ± 5.2	17.97 ± 2.3 *	20.76 ± 2.9	18.42 ± 3.5	21.52 ± 3.7	20.83 ± 4.1
liver							
	NEFAS [mmol/L]	6.97 ± 0.33	6.46 ± 0.42	7.14 ± 0.54	7.26 ± 0.28	7.48 ± 0.24	6.85 ± 0.75
	cholesterol/HDL-cholesterol ratio	7.73 ± 0.36	8.05 ± 0.37	8.50 ± 0.38	7.33 ± 0.31	8.18 ± 0.52	7.50 ± 0.58
	lactate [mmol/L]	0.44 ± 0.03	0.47 ± 0.03	0.65 ± 0.04 *	0.82 ± 0.05	0.81 ± 0.04	0.77 ± 0.05
	pyruvate [mmol/L]	0.41 ± 0.03	0.48 ± 0.04	0.57 ± 0.05 *	0.52 ± 0.09	0.56 ± 0.06	0.48 ± 0.05
serum							
	NEFAS [mmol/L]	0.61 ± 0.05	0.69 ± 0.05	0.64 ± 0.06	0.70 ± 0.08	0.69 ± 0.09	0.63 ± 0.04
	cholesterol/HDL-cholesterol ratio	3.39 ± 0.07	3.27 ± 0.07	3.86 ± 0.13 *	3.18 ± 0.08	3.07 ± 0.09	3.45 ± 1.0
	LDL-cholesterol/HDL ratio-cholesterol	1.81 ± 0.06	1.73 ± 0.11	2.28 ± 0.11 *	1.90 ± 0.09	1.77 ± 0.10	2.16 ± 0.10
	fasting glucose [ng/µL]	47.5 ± 5.44	64.1 ± 4.58	57.2 ± 3.81	34.4 ± 1.59	40.3 ± 5.45	45.4 ± 2.99 **
	lactate [mmol/L]	2.34 ± 0.22	2.48 ± 0.16	2.01 ± 0.13	2.92 ± 0.16	2.81 ± 0.14	2.96 ± 0.17
	estrogen [pg/mL]				132.6 ± 5.0	140.1 ± 16.8	130.8 ± 7.8

Data presented as group mean + SE. * denotes significance of *p* < 0.05, ** denotes significance of *p* < 0.005 with *n* = 10 animals per group. The statistical significance of the treated groups refers to the respective control group. NEFAs = non-esterified fatty acids; HDL = high density lipoprotein; LDL = low density lipoprotein; m = male; f = female; Contr = controls; Halo = haloperidol-medicated; Cloz = clozapine-medicated.

## Data Availability

The datasets used and/or analyzed during the current study are available from the corresponding author on reasonable request.

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
