# Peer review of "Sex-Specific Effects of Long-Term Antipsychotic Drug Treatment on Adipocyte Tissue and the Crosstalk to Liver and Brain in Rats"

_ijms, 2024, doi:10.3390/ijms25042188_

Round 1

Reviewer 1 Report (Previous Reviewer 2)

Comments and Suggestions for Authors

In the revised version of the manuscript, the authors have addressed all my remarks. Nevertheless, the quality of the data presentation is still insufficient and needs to be improved:

1. Please provide Table 1 and Supplementary Figure 3 at higher resolution.

2. In Supplementary Figure 1 (top blot), the perilipin-A bands are highlighted, as requested. However, there are two arrows indicating 50 and 75 kDa, which do not match the highlighted area. Please correct the positioning of the arrows (and delete one of them, if appropriate).

3. Due to the low resolution of Supplementary Figure 3 and its legend, it is not clear if the meaning of the asterisks was included in the figure legend (as requested in previous remark #6). Please provide the legend of this figure as text (instead of image) and confirm that it contains the asterisks.

Author Response

Reviewer 2 Report (New Reviewer)

Comments and Suggestions for Authors

The paper submitted to IJMS reports the effects of chronic exposure to two antipsychotics in male and female rats. I find this investigation interesting, and I particularly like including both sexes; there is a large body of literature using just females, who tend to gain more weight following antipsychotic treatment. This study is a part of a larger project with data published before.

I have some minor remarks, see below.

1. Previously published data. I am not sure how different is the dataset used for the Figure 1 from the previous papers. If it is re-analyzed, please, indicate how. I did not find the information. Maybe you could explain that the experimental animals are the same as in the previous studies in the last paragraph of the introduction? Or anywhere you find it meaningful.

2. Methods. Animals – did you order sibling on purpose? Why not unrelated rats? You claim they were housed in groups of 6. Can you specify the cage? Or you just mean the transport boxes? During individual housing, did you use enrichment?

3. Terminology: Please, consider using sex rather than gender, which is a social concept. I am not very comfortable with “medication”, I prefer treatment myself, but I do not insist.

4. Graphs: Please, remove the grey background and increase the resolution. This applies to the table 1 and figure 4 too. For the M/F distinction you may consider color coding such as blue/red or similar.

5. Table 1 legend: Do I understand correctly that the significance is in comparison to the respective (M or F) control group? Please, specify it in the legend.

Author Response

This manuscript is a resubmission of an earlier submission. The following is a list of the peer review reports and author responses from that submission.

Round 1

Reviewer 1 Report

Comments and Suggestions for Authors

The present manuscript hopes to clarify the sex-specific changes in the brain-liver-fat crosstalk under antipsychotic drug medication in rats. It sounds an interested topic. But the manuscript should be a research report based on present results than review-like paper based on the present results and the previous published results. It seems a logical mess at the moment. So, the manuscript needs to be overall arrangement.  In addition, present tables and figures need to be better designed: Table 1 should be designed as general three-line table; Figure1 didn’t show stained and calculated results; Figure2 should show the typical stained photographs and the calculated histograms; and the title and notes needs to be improved so as to increase their self-evident and normalization. CONCLUSION should only result from the present experiment.

Comments on the Quality of English Language

The quality of English language in the manuscript needs to be improved. I.e., The abstract can be presented by the logical relationships of purpose, method, result, and conclusion, and deleted the results and statements not covered in this manuscript such as AMPK, the aryl hydrocarbon receptor (AhR) …. The items in Results and Methods sections needs to be more precise expression: i.e “2.3. Glucose, glycogen, lactate “ can be “2.3. Effect of haloperidol or clozapine on glucose, glycogen and lactate contents in rat perirenal fat” …… “4.3. Neutral fat content and adipocyte area value” can be “4.3. Determination of neutral fat content and adipocyte area value” …… In addition,  many word expression such as “the pperirenal fat”, “some drug-induced physiological mechanisms underlying both conditions” , “Total fat, TGs and NEFAs in visceral adipocytes were not significantly increased in contrast to liver mass”, “Activity and muscle mass did not change under clozapine treatment”, “These findings contrast markedly with the data reported here and by Cooper et al. 2008 [41]” .…..  needs to be carefully considered. And there are many other issues on writting 

Author Response

We like to thank the reviewers for the constructive criticism, that really improves the quality of our manuscript. We have considered each single comment and have made the appropriate changes of the manuscript in response to the comments. All content changes in the revised manuscript have been marked in yellow.

Response to reviewer`s comments

Reviewer 1:

…..present tables and figures need to be better designed: Table 1 should be designed as general three-line table

Tab.1 is now redesigned by the editorial office. Now we added helpful information in the legends of figures and the table and improved the supplementary figures.

Figure1 didn’t show stained and calculated results

In the legend of Fig. 1 we mentioned the staining with Mayer’s hematoxylin. The calculated results are given in Tab.1 as “adipocytes area values` as stated in the legend of Fig.1.

Figure2 should show the typical stained photographs and the calculated histograms

After consultation with the editorial office we present the Western blotting results as histograms in Fig.2 and show the corresponding audioradiographs of the whole gels in Fig.1 suppl..

…the title and notes needs to be improved so as to increase their self-evident and normalization.

The title was changed as recommended by the Editor and the legends are more detailed now.

CONCLUSION should only result from the present experiment.

We omitted the outlook for further studies. Instead we inserted more results of the present study (page 12).

The quality of English language in the manuscript needs to be improved.

English language was improved by an English teacher.

The abstract can be presented by the logical relationships of purpose, method, result, and conclusion, and deleted the results and statements not covered in this manuscript such as AMPK, the aryl hydrocarbon receptor (AhR)

Although not explicitely stated, the abstract is structured in purpose, results and conclusion. To further strenghthen the purpose we inserted ` The aim of the present study was to investigate the underlying pathways of the metabolic dysfunctions.` Statements not covered by this study were deleted in the abstract.

The items in Results and Methods sections needs to be more precise expression

We improved the titles of 2.1-2.5 and 4.3-4.8  in order to make them more self-explanatory (highlighted in yellow)

In addition,  many word expression such as “the pperirenal fat”, “some drug-induced physiological mechanisms underlying both conditions” , “Total fat, TGs and NEFAs in visceral adipocytes were not significantly increased in contrast to liver mass”, “Activity and muscle mass did not change under clozapine treatment”, “These findings contrast markedly with the data reported here and by Cooper et al. 2008 [41]” .…..  needs to be carefully considered.

We changed the following word expressions

Pperirenal fat – perirenal fat (Abstract)

some drug-induced physiological mechanisms underlying both conditions - This unexpected link suggests that the drug-induced changes affect the lipid metabolism.( page 2 lane 2-3).

Total fat, TGs and NEFAs in visceral adipocytes were not significantly increased in contrast to liver mass  -  … the amount of fat and the level of TGs and NEFAs in visceral adipocytes were not significantly increased. In contrast, liver mass significantly changed under antipsychotic drug treatment [40] (page 8 lane 7-9).

Activity and muscle mass did not change under clozapine treatment  -  Neither a decrease in physical activity nor a reduction of muscle mass was observed under clozapine treatment [37;41] (page 8 lane 11-13).

These findings contrast markedly with the data reported here and by Cooper et al. 2008 [41]  - These findings contrast markedly to the data reported here and by Cooper et al. [41]. The difference might partially be explained by the time of drug administration. Sondhi et al. applied clozapine by gavage in the evening and observed sedation during the active night time. In addition, blood was taken from the animals without prior fasting, while the blood samples of our rats were taken 12 hours after food removal - thereby revealing true fasting glucose and hormone levels (page 8 lane 22-26).    

Reviewer 2 Report

Comments and Suggestions for Authors

In the present manuscript, Fehsel and Bouvier tested the metabolic effects of the antipsychotic drugs haloperidol and clozapine on rats. Both drugs affected glucose and lipid metabolism; however, some effects were sex-dependent. For example, clozapine administration increased fat mass only in male rats. In Figure 3, the conclusions of the present work were integrated with related findings from the literature.

This research could be useful in understanding any different responses in female and male patients to these drugs.

Minor remarks:

1. In the abstract, please correct “pperirenal” to “perirenal”.

2. In Figure 1, many of the adipocytes shown seem to be marked with crosses. Please indicate the meaning of these crosses in the respective figure legend.

3. In Table 1, please include a footer expanding the abbreviations used in this table.

4. In Supplementary Figure 1, please indicate the bands of interest with a rectangular red frame. It is suggested to keep the arrows and the molecular weight labels.

5. In supplementary figure 2, please label the axes in all graphs.

6. In the legend of Supplementary Figure 3, please indicate the meaning of the asterisks and include the full names of the abbreviations used in these graphs.

7. Please consider removing "ferric iron" from the title of section 2.1, since no ferric iron quantification results are shown.

8. In section 2.4, please indicate the figure or table containing the data.

9. In the discussion, please replace the "APDs" acronym with its full name, or expand this acronym when using it for the first time.

10. The end of the abstract mentions the activation of the glycerol shunt and AMPK; however, these hypotheses were not tested here. As such, please consider drawing a conclusion based only on experimental findings from the present research.

Round 2

Reviewer 1 Report

Comments and Suggestions for Authors

Although the manuscript has been improved based on previous version, it needs much improvement. 

1.        Based on the present results, its title might be changed as "Sex-specific changes in the lipid metabolism under antipsychotic drug medication in rats”

2.       The conclusion in abstract was not from the present results. How to get the conclusion that “Thus, our clozapine-feeded rats are an attractive animal model for studies on age- , sex- or drug-related metabolic disorders like insulin resistance, weight gain and hepatic steatosis”? In addition, there might be grammatical problems on “although they present with hyperglycemia not counter-regulated by elevated insulin secretion.”

3.       “Introduction” section, why had the difference on “AhR” and “AHR”?

4.       Table 1 in Results sections might be changed 4 tables so as to fit the precise expression of 2.1 – 2.4 be more precise expression, and the words expression of the result based on tables needs to be improved to be understanded on comparison between groups.  The words “The values marked in grey are significant compared to controls with *p<0.05, **p<0.001 and ***p<0.0001 as significant. NEFAS=non-esterified fatty acids; TyG-Index= triglyceride-glucose-index; HDL= high density lipoprotein; LDL= low density lipoprotein; BW=body weight.” should be note of the table than title of the table, and should be below the table.

5.       DISCUSSION section: Many words need to be carefully considered such as “did not significantly contribute to the weight changes under antipsychotic drug treatment in our rats”, “Weight gain of male clozapine treated rats or slowdown of weight gain of male and female haloperidol medicated rats persisted after subtraction of perirenal fat plus liver mass from final body weight”, “These findings contrast markedly to the data reported here and by Cooper et”, “The quite different physiological effects of haloperidol and clozapine on lipogenesis, lipolysis and sex can be explained by the degree of their AhR activation”, “Increased expression of the canonical AhR target gene Cyp1A1 was still detectable in human adipocytes after 14 days of clozapine treatment [19], but within 3 months of clozapine feeding hepatic Cyp1A1 protein levels had returned to baseline”, “but further characteristics of hypertrophic adipocytes like increased insulin or leptin plasma levels are missing”, .…. To make a long story short, readability needs further improvement. In addition, “Figure 3. Schematic diagram of the brain-liver-fat cross talk under clozapine medication” might be changed as “Figure 3. Schematic diagram of the lipid metabolism under clozapine medication” based present results in the manuscript so as to clarify the significance of the present results.

6.       CONCLUSION needs to be better condensed based on results and discussion.

7.       Methods section: The design includes 6 groups. Why “housed in 5 groups”?

Comments on the Quality of English Language

Although the manuscript has been improved based on previous version, it needs much improvement. 

1.        Based on the present results, its title might be changed as "Sex-specific changes in the lipid metabolism under antipsychotic drug medication in rats”

2.       The conclusion in abstract was not from the present results. How to get the conclusion that “Thus, our clozapine-feeded rats are an attractive animal model for studies on age- , sex- or drug-related metabolic disorders like insulin resistance, weight gain and hepatic steatosis”? In addition, there might be grammatical problems on “although they present with hyperglycemia not counter-regulated by elevated insulin secretion.”

3.       “Introduction” section, why had the difference on “AhR” and “AHR”?

4.       Table 1 in Results sections might be changed 4 tables so as to fit the precise expression of 2.1 – 2.4 be more precise expression, and the words expression of the result based on tables needs to be improved to be understanded on comparison between groups.  The words “The values marked in grey are significant compared to controls with *p<0.05, **p<0.001 and ***p<0.0001 as significant. NEFAS=non-esterified fatty acids; TyG-Index= triglyceride-glucose-index; HDL= high density lipoprotein; LDL= low density lipoprotein; BW=body weight.” should be note of the table than title of the table, and should be below the table.

5.       DISCUSSION section: Many words need to be carefully considered such as “did not significantly contribute to the weight changes under antipsychotic drug treatment in our rats”, “Weight gain of male clozapine treated rats or slowdown of weight gain of male and female haloperidol medicated rats persisted after subtraction of perirenal fat plus liver mass from final body weight”, “These findings contrast markedly to the data reported here and by Cooper et”, “The quite different physiological effects of haloperidol and clozapine on lipogenesis, lipolysis and sex can be explained by the degree of their AhR activation”, “Increased expression of the canonical AhR target gene Cyp1A1 was still detectable in human adipocytes after 14 days of clozapine treatment [19], but within 3 months of clozapine feeding hepatic Cyp1A1 protein levels had returned to baseline”, “but further characteristics of hypertrophic adipocytes like increased insulin or leptin plasma levels are missing”, .…. To make a long story short, readability needs further improvement. In addition, “Figure 3. Schematic diagram of the brain-liver-fat cross talk under clozapine medication” might be changed as “Figure 3. Schematic diagram of the lipid metabolism under clozapine medication” based present results in the manuscript so as to clarify the significance of the present results.

6.       CONCLUSION needs to be better condensed based on results and discussion.

7.       Methods section: The design includes 6 groups. Why “housed in 5 groups”?

Author Response

We like to thank the reviewers for the constructive criticism, that really improves the quality of our manuscript. We have considered each single comment and have made the appropriate changes of the manuscript in response to the comments. All content changes in the revised manuscript have been marked in turquoise colour.

Response to reviewer`s comments

1.Based on the present results, its title might be changed as "Sex-specific changes in the lipid metabolism under antipsychotic drug medication in rats”

We feel that the title proposed by the reviewer is not adequate in regard to the message of this study. We present results of liver, serum, perirenal fat (for the first time) and brain (former study) for each individual rat after 12 weeks of drug feeding. Our study aimed to build a physiologically based model reflecting observed changes in physiological and molecular parameters relevant to the well-known side effects of antipsychotic treatment. On page 13 lane 16, we inserted the blood collection and the dissection of liver to emphasise the contribution of liver and serum parameters to our model of metabolic disorder.

2a.The conclusion in abstract was not from the present results. How to get the conclusion that “Thus, our clozapine-feeded rats are an attractive animal model for studies on age- , sex- or drug-related metabolic disorders like insulin resistance, weight gain and hepatic steatosis”?

Our model reflects the incidences of the metabolic syndrome (hyperglycemia, weight gain and insulin resistance) as well as hepatic steatosis. Grant et al (World J Gastroenterol 2022) revealed that antipsychotics and metabolic syndrome are contributing factors to liver disease. In line with these results, M Gunther and JA Dopheide (2023) confirmed that chlorpromazine, clozapine, and olanzapine pose the greatest risk of hepatotoxicity in patients.

Consistently with previous studies, our data confirm the presence of sex-related differences in clozapine tolerability, with effects of sex especially on weight gain, dyslipidemia and hyperglycemia. Although non-life-threatening, these common adverse effects significantly affect patients' quality of life, undermine compliance, and might cause treatment discontinuation. This study contributes to a better understanding of this topic and might contribute to tailor therapeutic approaches, thus improving tolerability, compliance and clinical stability. Finally, we remain of the view that our rat model is well suited to study age- , sex- or drug-related metabolic disorders like insulin resistance, weight gain and hepatic steatosis.

2b.       In addition, there might be grammatical problems on “although they present with hyperglycemia not counter-regulated by elevated insulin secretion.”

We changed this part of the sentence - `hyperglycemia without hyperinsulinemia` in the abstract.

  1. “Introduction” section, why had the difference on “AhR” and “AHR”?

We corrected the mistake on page 2 lane 38.

  1. Table 1 in Results sections might be changed 4 tables so as to fit the precise expression of 2.1 – 2.4 be more precise expression, and the words expression of the result based on tables needs to be improved to be understanded on comparison between groups. The words “The values marked in grey are significant compared to controls with *p<0.05, **p<0.001 and ***p<0.0001 as significant. NEFAS=non-esterified fatty acids; TyG-Index= triglyceride-glucose-index; HDL= high density lipoprotein; LDL= low density lipoprotein; BW=body weight.” should be note of the table than title of the table, and should be below the table.

We revised Tab.1. Instead of creating 4 very small tables, we separated the results of the different organs by blank lines to make the presentation more clearly. We now created a food note of the text that was originally part of the title.

Unfortunaly, we don`t understand the meaning of the part of the sentence ` and the words expression of the result based on tables needs to be improved to be understanded on comparison between groups.`

  1. DISCUSSION section: Many words need to be carefully considered such as “did not significantly contribute to the weight changes under antipsychotic drug treatment in our rats”

We think you criticized the word order in the sentence.

We changed it on page 8 lane 3.

“Weight gain of male clozapine treated rats or slowdown of weight gain of male and female haloperidol medicated rats persisted after subtraction of perirenal fat plus liver mass from final body weight”,

We replaced `persisted` by `remained` on page 8 lane 9

 “These findings contrast markedly to the data reported here and by Cooper et”,

 We changed to :

“These findings markedly contrast with the data reported here and by Cooper et.al `

Page 8, lane 21

 “The quite different physiological effects of haloperidol and clozapine on lipogenesis, lipolysis and sex can be explained by the degree of their AhR activation”,

We omitted `their` on page 8, lane 28

 “Increased expression of the canonical AhR target gene Cyp1A1 was still detectable in human adipocytes after 14 days of clozapine treatment [19], but within 3 months of clozapine feeding hepatic Cyp1A1 protein levels had returned to baseline”

We changed to:

“The canonical AhR target gene Cyp1A1 was strongly expressed in human adipocytes after 14 days of clozapine treatment [19], while the hepatic Cyp1A1 protein level was not increased after 3 months of clozapine feeding [40].” Page 8, page 31-33

“but further characteristics of hypertrophic adipocytes like increased insulin or leptin plasma levels are missing”.

We changed to:

but further characteristics like hypertrophic adipocytes as well as increased insulin or leptin plasma levels are missing. Page 8, lane 36

 “Figure 3. Schematic diagram of the brain-liver-fat cross talk under clozapine medication” might be changed as “Figure 3. Schematic diagram of the lipid metabolism under clozapine medication” based present results in the manuscript so as to clarify the significance of the present results.

As suggested by the reviewer, we changed the title of Fig.3 but included glucose metabolism in the legend of Fig.3.

  1. CONCLUSION needs to be better condensed based on results and discussion.

The conclusion was shortened  - see page 12.

  1. Methods section: The design includes 6 groups. Why “housed in 5 groups”?

In the 5 groups the siblings were housed together during acclimatization before treatment. During treatment the rats were housed separately.

English language was improved in some sentences marked in green.

Round 3

Reviewer 1 Report

Comments and Suggestions for Authors

1.I thought that based on the present results than previous published, its title might be changed as "Sex-specific changes in the lipid metabolism under antipsychotic drug medication in rats”. The title of present manuscript should only be from the results of present experiments (mainly involved in liver, serum, perirenal lipids metabolism) than from the published data.

2.The conclusion in abstract was not from the results of present experiment. From the results, how to get that “clozapine-feeded rats are an attractive animal model”? And some sentences should be improved, such as The aim of the present study was to investigate the underlying pathways of the metabolic dysfunctions” should be“The aim of the present study was to investigate the underlying pathways of the metabolic dysfunctions caused by antipsychotic treatment”; “Adipocyte hypertrophy, iron content, proteins and metabolites of the glucose-/lipid-metabolism were studied.” Should be “Adipocyte hypertrophy, iron content, proteins and metabolites of the glucose-/lipid-metabolism in treated rats were determined”; “While haloperidol induced the wasting syndrome in the rats of both sexes, clozapine increased weight and fat mass only in males” should be “Haloperidol induced the wasting syndrome in the rats of both sexes, whereas clozapine increased weight and fat mass only in males”; “Insulin receptor ß (IRß) expression changed sex-specifically” also is not accurate, might be “Clozapine or haloperidol induced Insulin receptor ß (IRß) expression in perirenal adipocytes was sex-specifically changed”; and subsequent all statements were not accurate, need to be re-organized.

3. The presentation of results is very messy; figures and tables lack of standardization; the words explanations of the result based on figures and tables were not accurate; figure notes and table notes were not clear.

4.“Figure 3. Schematic diagram should be based on the results of present experiment than previous published data. The present manuscript is research file than review file.

5. English language still has some flaws such as “This is achieved by crosstalk between fat depots and the brain”, “dyslipidemia. [14]”, “Male clozapine treated animals”, “female haloperidol medicated rats”, …….  needs to more authentic editions..

Comments on the Quality of English Language

1.I thought that based on the present results than previous published, its title might be changed as "Sex-specific changes in the lipid metabolism under antipsychotic drug medication in rats”. The title of present manuscript should only be from the results of present experiments (mainly involved in liver, serum, perirenal lipids metabolism) than from the published data.

2.The conclusion in abstract was not from the results of present experiment. From the results, how to get that “clozapine-feeded rats are an attractive animal model”? And some sentences should be improved, such as The aim of the present study was to investigate the underlying pathways of the metabolic dysfunctions” should be“The aim of the present study was to investigate the underlying pathways of the metabolic dysfunctions caused by antipsychotic treatment”; “Adipocyte hypertrophy, iron content, proteins and metabolites of the glucose-/lipid-metabolism were studied.” Should be “Adipocyte hypertrophy, iron content, proteins and metabolites of the glucose-/lipid-metabolism in treated rats were determined”; “While haloperidol induced the wasting syndrome in the rats of both sexes, clozapine increased weight and fat mass only in males” should be “Haloperidol induced the wasting syndrome in the rats of both sexes, whereas clozapine increased weight and fat mass only in males”; “Insulin receptor ß (IRß) expression changed sex-specifically” also is not accurate, might be “Clozapine or haloperidol induced Insulin receptor ß (IRß) expression in perirenal adipocytes was sex-specifically changed”; and subsequent all statements were not accurate, need to be re-organized.

3. The presentation of results is very messy; figures and tables lack of standardization; the words explanations of the result based on figures and tables were not accurate; figure notes and table notes were not clear.

4.“Figure 3. Schematic diagram should be based on the results of present experiment than previous published data. The present manuscript is research file than review file.

5. English language still has some flaws such as “This is achieved by crosstalk between fat depots and the brain”, “dyslipidemia. [14]”, “Male clozapine treated animals”, “female haloperidol medicated rats”, …….  needs to more authentic editions..

Author Response

We have considered each single comment and have made the appropriate changes of the manuscript in response to the comments. All content changes in the revised manuscript have been marked in yellow colour.

Response to reviewer`s comments (round 3)

1.I thought that based on the present results than previous published, its title might be changed as "Sex-specific changes in the lipid metabolism under antipsychotic drug medication in rats”. The title of present manuscript should only be from the results of present experiments (mainly involved in liver, serum, perirenal lipids metabolism) than from the published data.

As explained in the previous round of revision, we think that the title proposed by the reviewer is not adequate in regard to the message of this study. We present results of liver, serum, perirenal fat (for the first time) and brain (former study) for each individual rat after 12 weeks of drug feeding. We now accommodated the wishes of the reviewer and deleted `brain` from the title.

2.The conclusion in abstract was not from the results of present experiment. From the results, how to get that “clozapine-feeded rats are an attractive animal model”?

As explained in the previous round of revision, our model reflects the incidences of the metabolic syndrome (hyperglycemia, weight gain and insulin resistance) as well as hepatic steatosis.

Consistently with previous studies, our data confirm the presence of sex-related differences in clozapine tolerability, with effects of sex especially on weight gain, dyslipidemia and hyperglycemia. This study contributes to a better understanding of this topic.  We now accommodated the wishes of the reviewer and changed the sentence : `Thus, clozapine-feeded rats are well suited to study age- , sex- or drug-related metabolic disorders like insulin resistance, weight gain and hepatic steatosis.`

some sentences should be improved, such as The aim of the present study was to investigate the underlying pathways of the metabolic dysfunctions” should be“The aim of the present study was to investigate the underlying pathways of the metabolic dysfunctions caused by antipsychotic treatment”;

We changed this sentence – page 1 lane 3

“Adipocyte hypertrophy, iron content, proteins and metabolites of the glucose-/lipid-metabolism were studied.” Should be “Adipocyte hypertrophy, iron content, proteins and metabolites of the glucose-/lipid-metabolism in treated rats were determined”;

We changed this sentence – page 1 lane 4

“While haloperidol induced the wasting syndrome in the rats of both sexes, clozapine increased weight and fat mass only in males” should be “Haloperidol induced the wasting syndrome in the rats of both sexes, whereas clozapine increased weight and fat mass only in males”

We changed this sentence – page 1 lane 6

Insulin receptor ß (IRß) expression changed sex-specifically” also is not accurate, might be “Clozapine or haloperidol induced Insulin receptor ß (IRß) expression in perirenal adipocytes was sex-specifically changed”; and subsequent all statements were not accurate, need to be re-organized.

We could not change this sentence as suggested by the reviewer, because clozapine and haloperidol do not induce insulin receptor ß expression; they just modulate its expression. However, we added ` by clozapine or haloperidol treatment`at the end of the sentence  - page 1 lane 8

subsequent all statements were not accurate, need to be re-organized.

We specified the other sentences in the abstract – marked in yellow.

The presentation of results is very messy; figures and tables lack of standardization; the words explanations of the result based on figures and tables were not accurate; figure notes and table notes were not clear.

We don`t understand the points of criticism in the Result section.

We tried to present our results as precise as possible. The style of this presentation resembles those in our other publications. They were accepted by former reviewers without objections.  

4.“Figure 3. Schematic diagram should be based on the results of present experiment than previous published data. The present manuscript is research file than review file.

The aim of the discussion is to integrate the new data into known physiological processes.

We are in the lucky position, that we can present additional former results from the same animals, that help to understand the influence of antipsychotic drug treatment on the metabolism of the whole body. In Fig.3 the new data are embedded in a holistic conception of changes induced by treatment.

  1. English language still has some flaws such as “This is achieved by crosstalk between fat depots and the brain”, “dyslipidemia. [14]”, “Male clozapine treated animals”, “female haloperidol medicated rats”, ……. needs to more authentic editions..

We  inserted `regulation` - “This regulation is achieved by crosstalk between fat depots and the brain” page 2 lane 8

We replaced  “dyslipidemia` by hypertriglyceridemia. [14]” page 2 lane 19.

 “Male clozapine treated animals”, “female haloperidol medicated rats”, …….  needs to more authentic editions.

The whole publication is about drug-feeded rats. Should we really substitute treated/ medicated by feeded in the whole publication?